# Production of Promising Heat-Labile Enterotoxin (LT) B Subunit-Based Self-Assembled Bioconjugate Nanovaccines against Infectious Diseases

**DOI:** 10.3390/vaccines12040347

**Published:** 2024-03-23

**Authors:** Caixia Li, Juntao Li, Peng Sun, Ting Li, Xue Yan, Jingqin Ye, Jun Wu, Li Zhu, Hengliang Wang, Chao Pan

**Affiliations:** State Key Laboratory of Pathogen and Biosecurity, Beijing Institute of Biotechnology, Beijing 100071, China; lcx19950630@163.com (C.L.); ljtanlz0046@163.com (J.L.); sunpeng990718@163.com (P.S.); liting7427@163.com (T.L.); cynthia2182@163.com (X.Y.); yejq0922@163.com (J.Y.); jewly54@bmi.ac.cn (L.Z.)

**Keywords:** LTB, bioconjugate nanovaccines, biosynthesis, glycosylation, self-assembling

## Abstract

Nanoparticles (NPs) have been widely utilized in vaccine design. Although numerous NPs have been explored, NPs with adjuvant effects on their own have rarely been reported. We produce a promising self-assembled NP by integrating the pentameric *Escherichia coli* heat-labile enterotoxin B subunit (LTB) (studied as a vaccine adjuvant) with a trimer-forming peptide. This fusion protein can self-assemble into the NP during expression, and polysaccharide antigens (OPS) are then loaded in vivo using glycosylation. We initially produced two *Salmonella paratyphi* A conjugate nanovaccines using two LTB subfamilies (LTIB and LTIIbB). After confirming their biosafety in mice, the data showed that both nanovaccines (NP(LTIB)-OPS_SPA_ and NP(LTIIbB)-OPS_SPA_) elicited strong polysaccharide-specific antibody responses, and NP(LTIB)-OPS resulted in better protection. Furthermore, polysaccharides derived from *Shigella* or *Klebsiella pneumoniae* were loaded onto NP(LTIB) and NP(LTIIbB). The animal experimental results indicated that LTIB, as a pentamer module, exhibited excellent protection against lethal infections. This effect was also consistent with that of the reported cholera toxin B subunit (CTB) modular NP in all three models. For the first time, we prepared a novel promising self-assembled NP based on LTIB. In summary, these results indicated that the LTB-based nanocarriers have the potential for broad applications, further expanding the library of self-assembled nanocarriers.

## 1. Introduction

Vaccines have significant importance for controlling and preventing infectious diseases. In bacteria, highly specific surface polysaccharides, including O-polysaccharide (OPS) and capsular polysaccharide (CPS), are considered ideal antigen targets [1]. Nevertheless, polysaccharides alone cannot stimulate effective protective immune responses due to the absence of T cell reactions. Conjugate vaccines, prepared by coupling bacterial polysaccharides with appropriate carrier proteins, convert polysaccharides from T cell-independent antigens into T cell-dependent antigens, thus triggering potent immune responses [2]. Conjugate vaccines are currently considered the most successful bacterial vaccines, and several products, including PCV13 and PedvaxHIB, have been launched. All of these marketed conjugate vaccines are synthesized through chemical methods. However, advances in synthetic biology and vaccine technology have resulted in an efficient new biosynthesis technology that remedies numerous disadvantages of traditional chemical methods [3,4].

Conjugate vaccine biosynthesis technology, also known as protein glycan coupling technology (PGCT), can directly synthesize glycoproteins in engineered bacteria via enzymatic reactions [5]. During the process of LPS synthesis, OPS is transferred to the lipid A-core under the catalysis of O-antigen ligase WaaL. Due to the similarity between LPS synthesis and protein glycosylation, the *waaL* gene could be knocked out in the host bacteria, blocking the binding of OPS to lipid A-core. Then, by co-expressing OPS, glycosyltransferase, and the carrier protein (with a glycosylation modification motif), the OPS will be transferred to the synthesized carrier under the catalysis of PglL, creating a glycoprotein. Generally, gene clusters that encode pathogenic bacterial polysaccharides, carrier proteins, and glycosyltransferases can be transformed into engineered *E. coli*. After being induced, the synthesized polysaccharides directly couple with the carrier proteins under the catalysis of glycosyltransferases. Therefore, glycoproteins with higher yields and better uniformity are produced using this method that only requires one-step catalysis and purification. In addition, the glycosylation system can be directly established within attenuated host bacteria, and this avoids the need to clone an extensive fragment of the polysaccharide gene cluster and the low-efficiency synthesis of heterologous polysaccharides in *E. coli* [6,7]. There are currently three useful glycosyltransferases for preparing conjugate vaccines that have been identified, namely, PglB, PglL, and PglS [8]. PglB was the first to be applied in the biosynthesis of conjugate vaccines [9,10,11], followed by PglL, thus serving a wider range of applications [12,13]. PglS was recently identified and can be used for the glycosylation of capsular polysaccharides [14].

Nanoparticles (NPs), showcasing superior targeting and immune-stimulating abilities, have been extensively used in vaccines following the development of nanotechnologies [15]. For prophylactic vaccines, proteinaceous NPs have received more attention for their greater safety and biocompatibility. As such, various virus-like particles, such as HBc, AP205, and Qβ, have been adopted in SARS-CoV2 vaccine research [16,17,18]. Additionally, ferritin-NPs, which can induce a receptor-mediated immune response, have been used to produce SARS-CoV-2, influenza, HIV-1, Epstein–Barr, and hepatitis C virus vaccines [19,20,21,22,23]. In addition to these natural monomer protein assemblies, a novel modular self-assembly design has been developed recently in which multiple oligomeric modules can form higher-order polymeric nanostructures in space [24,25]. Utilizing these self-assembly principles, we fused the cholera toxin B subunit (CTB), a well-known mucosal adjuvant, with an artificially designed trimer-forming peptide (Tri) and self-assembled this fusion into NPs in cells [26]. Subsequently, by combining with PGCT, we produced bacterial bioconjugate nanovaccines with robust humoral and cellular immune responses [26,27]. Moreover, these nanoparticles have been used to prepare an inhalable COVID-19 vaccine, resulting in a potent mucosal immune response [28]. These studies indicate that the CTB-based self-assembled NP can stimulate robust immune responses and provide effective protection. This pentamer, CTB, is not only a key module for self-assembly, but it also exhibits an adjuvant function; thus, it has greater potential for applications. In actuality, the family of bacterial AB5 toxins also comprises an array of members that include the heat-labile enterotoxin (LT), Shiga toxin (Stx), and subtilase cytotoxin (SubAB) [29].

LT is produced by enterotoxigenic *E. coli*. In addition, both LT and CT belong to the cholera toxin family, one of four AB5 toxin families [29]. Furthermore, the LT toxin can be subdivided into two subfamilies of type I (LTI) and type II (LTII), and the latter also can be divided into LTIIa and LTIIb. The B-subunits of CT, LTI, and LTII can assemble to form a homo-pentameric, ring-shaped scaffold, although they have a low level of sequence identity. It has also been reported that similar to CTB, the B-subunit of the LTI (LTIB) pentamer can specifically recognize the monosialotetrahexosylganglioside (GM1), while LTIIaB and LTIIbB show affinity for the gangliosides GD1b and GD1a, respectively [30,31].

In this study, we prepared two self-assembled NPs by fusing pentameric LTB (LTIB or LTIIbB) and a trimer-forming peptide. We first established a PglL-based glycosylation system in *S. paratyphi* A and prepared two bioconjugate nanovaccines, (NP(LTIB)-OPS_SPA_ and NP(LTIIbB)-OPS_SPA_). After confirming their safety and great antibody responses, the powerful protective effect of NP(LTIB)-OPS was further determined. To achieve the widespread application of this vaccine, we subsequently coupled the polysaccharides derived from *Shigella* or *K. pneumoniae* to NP(LTIB) and NP(LTIIbB), respectively. The results from the animal experiments demonstrated that the most potent immune response and protection was showcased with NP(LTIB)-OPS for both strains, consistent with the results of NP(CTB)-OPS. Therefore, we provided a novel promising self-assembled NP and further expanded the application of multimodule self-assembled nanocarriers.

## 2. Materials and Methods

### 2.1. Strain, Plasmid, and Growth Conditions

50973DWC/CldLT2 was a mutation of *S. paratyphi* A CMCC strain 50973 in which the *waaL* and *cld* genes were removed and the polysaccharide chain extension gene *cld* from *Salmonella typhimurium* was introduced. The 301DWP was a mutation *S. flexneri* 2a strain 301 in which the *waaL* was deleted, and the virulent plasmid, pCP, was removed. 355DW was a strain of *K. pneumoniae* strain 355 with the *waaL* gene deleted. All of these strains have been previously detailed in our work [6,7,32]. The plasmids pPglL-LTIBTri and pPglL-LTIIbBTri were developed by replacing CTB with LTIB and LTIIbB, respectively, based on pPglL-CTBTri, which has been described in our prior studies. To construct the expression strains, these plasmids were electrotransferred into the aforementioned bacteria. The bacteria were cultivated in either a liquid or solid lysogeny broth (LB) medium with 1.5% agar. For the glycoprotein expression, 1 mM isopropyl β-d-1-thiogalactopyranoside (IPTG) was added when the cells were cultured until optical density (OD) = 0.6–0.8 at 37 °C, and then cultured at 30 °C for another 10 to 12 h.

### 2.2. Coomassie Blue Staining and Western Blotting

After centrifugation, the cultured cells were resuspended using ddH_2_O. The samples were then added to an equal volume of 2× SDS buffer consisting of 100 mM Tris-HCl (pH 6.8), 3.2% (*w*/*v*) sodium dodecyl sulfate (SDS), 0.04% (*w*/*v*), bromophenol blue dye, 16% (*v*/*v*) glycerol, and 40 mM DL-dithiothreitol. They were then placed in a boiling bath for 10 min. Following this, 20 µL of the sample was separated using sodium dodecyl sulfate and polyacrylamide gel (SDS-PAGE) and stained with Coomassie blue or transferred to a nitrocellulose membrane. The membrane was then blocked with a blocking solution, tris-buffered saline and Polysorbate 20 (TBST) that contained 5% skim milk powder, for 1 h at 37 °C. After washing three times with TBST, HRP-conjugated anti-6×His tag antibodies (Abmart, Shanghai, China, M20020L) were subsequently added and incubated at 37 °C for 1 h. After another washing step, the bands were visualized with the color developing solution (Thermo Fisher Scientific, Waltham, MA, USA 32106) using a Tanon 5200 imaging system. For the detection of *S. paratyphi* A CMCC strain 50,973 and *S. flexneri* 2a strain 301 polysaccharides, OPS-specific serums (Denka Seiken, Tokyo, Japan, 211255, 210227) were used for the primary antibodies and HRP-conjugated anti-rabbit IgG (Transgen Biotech, Beijing, China, HS101-01) as the secondary antibodies. Serum against the *K. pneumoniae* strain 355 polysaccharide was used as described above [27].

### 2.3. LPS and OPS Preparation

The lipopolysaccharides (LPS) were extracted using the hot phenol method, and the details are as follows. The wild strain was cultured for 12 h and then centrifuged at 8000 rpm for 10 min to collect the supernatant. This was then washed three times with pre-cooled ddH_2_O and finally resuspended in ddH_2_O. An equal volume of 90% phenol solution was added and centrifuged at 68 °C for 30 min at 8000 rpm. The upper aqueous phase was aspirated, and the water extraction was repeated. All of the aqueous phases were transferred to a dialysis bag with a molecular weight cut-off of 3.5 kDa for 3 days. DNase and RNase were added for reaction at 37 °C for 4 h, then Proteinase K was added for reaction at 60 °C for 4 h, and, finally, they were placed in a boiling water bath for 10 min. The supernatant was collected as the LPS solution. Glacial acetic acid was added to the LPS solution for a final concentration of 1% and boiled for 90 min. We then used 0.5 M/mL NaOH solution to adjust the pH to 7.0. Finally, this was centrifuged at 40,000× *g* for 5 h, and the supernatant was the OPS solution.

### 2.4. Glycoprotein Expression and Purification

Post-expression, cells were gathered via centrifugation at 8000 rpm and subsequently resuspended using Solution A1, containing 20 mM Tris-HCl (pH 7.5), 10 mM imidazole, and 500 mM NaCl. These resuspended cells were homogenized using a high-pressure homogenizer (Ph.D. Technology LLC, Saint Paul, MN, USA, D-6L). After another round of centrifugation, the resulting supernatant was enriched by passage through a nickel affinity chromatography column and later eluted with Solution B1 containing 20 mM Tris-HCl (pH 7.5), 500 mM imidazole, and 500 mM NaCl. The eluted solution was concentrated via a 10 kDa ultrafiltration cube. Subsequently, this was passed through a column filled with 24 mL of Superdex 200 with phosphate-buffered solution (PBS) as the mobile phase at a flow rate of 1 mL/min. The isolated samples were collected and analyzed using Coomassie blue staining.

### 2.5. Experimental Animals

Pathogen-free female BALB/c mice (6–8 weeks old) were purchased from SPF Biotechnology (Beijing, China) and housed at the Laboratory Animal Centre of the Academy of Military Medical Sciences. All of the animal experiments were approved by the Academy of Military Medical Sciences Animal Care and Use Committee (Ethics Approval Code IACUC-DWZX-2020-044, approval date 2020.04.30).

### 2.6. Safety Estimation

The mice were immunized subcutaneously with NP(LTIB)-OPS or NP(LTIIbB)-OPS at doses of 25 μg of polysaccharide diluted in 100 μL of PBS, which is 10 times the conventional immunization dose. Body temperatures and weights were measured on days 0, 0.5, 1, 2, 5, 10, and 15, and tail blood was collected at the same time to detect the concentration of interleukin-6 (IL-6), IL-1β, tumor necrosis factor-α (TNF-α), and interferon-γ (IFN-γ) cytokines. On day 30, aspartate aminotransferase (AST), alanine aminotransferase (ALT), alkaline phosphatase (ALP), lactate dehydrogenase (LDH), and blood urea nitrogen (BUN) in the serums were analyzed using an automated Hitachi-917 analyzer. In addition, tissue sections from the hearts, livers, spleens, lungs, and kidneys were stained with hematoxylin and eosin (H&E) for analysis.

### 2.7. Immunization Experiments

The mice were randomly divided into five groups, namely, PBS, OPS, NP(CTB)-OPS, NP(LTIB)-OPS, and NP(LTIIbB)-OPS. Apart from the PBS group, every mouse in the other groups was injected subcutaneously on days 0, 14, and 28 with each type of vaccine containing 2.5 μg of polysaccharide diluted in PBS with no adjuvant. Following a one-week period after each immunization, blood samples were harvested from the tail tips, and the serums were subsequently isolated and stored at −80 °C for future analysis. Antibody titers in the serums against the pathogen LPS were measured using an enzyme-linked immunosorbent assay (ELISA). At 14 days following the third immunization, the mice were intraperitoneally injected with cultured bacteria for challenge tests to assess the protective efficacy of the vaccine.

### 2.8. Enzyme-Linked Immunosorbent Assay (ELISA)

The 96-well plates were coated with 10 μg/well of LPS diluted in a coating solution comprising 50 mM of Na_2_CO_3_-NaHCO_3_ (pH 9.6) overnight at 4 °C. Following three washings with phosphate-buffered saline (PBST) and subsequent drying, each well had 200 μL of a blocking solution added, and they were incubated for 2 h at 37 °C. After an additional washing step, diluted immune serums (100 μL/well in dilution buffer; PBST with 0.5% skimmed milk powder) were added, and the plates were incubated at 37 °C for 1 h. After washing and drying again, each well had 100 μL of HRP-conjugated goat anti-mouse antibodies (diluted 1:10,000 in dilution buffer) added that included IgG, IgG1, IgG2a, IgG2b, and IgG3 (Abcam, Cambridge, UK, AB6820, ab97240, ab97245, ab97250, ab97260). The plates were incubated for 1 h at 37 °C. Post-washing, each well was treated with 100 μL of TMB solution for 10 min at room temperature. The reaction was stopped by the addition of 50 mL/well of a termination solution (2M H_2_SO_4_). A microplate spectrophotometer was then utilized to read the plate at an optical wavelength of 450 nm.

### 2.9. Statistical Analysis

Data are presented as means ± s.d., and the statistical analysis was performed in GraphPad Prism 8.0 software using one-way analysis of variance (ANOVA) with Dunnett’s multiple comparison test for the multiple-group comparison. Statistically significant differences are indicated as * *p* < 0.05, ** *p* < 0.01, *** *p* < 0.001, **** *p* < 0.0001.

## 3. Results

### 3.1. Production of the S. paratyphi A Self-Assembled Bioconjugate Nanovaccine with LTBs as the Pentamer Module

In a prior experiment, an NP was produced by fusing CTB with a trimeric peptide, Tri, and it exhibited its potential by eliciting strong humoral, cellular, and mucosal immune responses [26,28]. Considering the important role of LTB in immunity, we substituted the sequence of CTB with LTIB and LTIIbB on the base of pPglL-CTBTri. Following this, the recombinant plasmids containing pPglL-LTIBTri and pPglL-LTIIbBTri were separately transformed into the host strain, 50973DWC/CldLT2, capable of expressing the long *S. paratyphi* A CMCC strain 50973 OPS chain (OPS_SPA_) (Figure 1A). After inducing with IPTG overnight, all of the cell lysis samples were separated using SDS-PAGE and subsequently identified by Coomassie blue staining and Western blotting using the HRP-labeled 6×His Tag antibody. It was distinctly observed that all of the fused proteins displayed typical ladder-like bands above each carrier protein, indicating that the polysaccharide antigens were coupled with carrier proteins through glycosylation (Figure 1B). Subsequently, the glycoproteins were purified using affinity and size-exclusion chromatography. During the size-exclusion chromatography step, it was noted that the eluted columns of both glycoproteins were approximately <10 mL (lower than 30% of the column volume), indicating polymer formation (Figure 1C,D). The Coomassie blue staining showed that the purity of the final samples (LTIBTri-OPS_SPA_ and LTIIbB-OPS_SPA_) exceeded 90%, and the Western blotting using an HRP-labeled 6×His Tag antibody and *S. paratyphi* A-specific O2 serum confirmed the successful linking of the *S. paratyphi* A CMCC strain 50973 OPS and carrier proteins (Figure 1E).

### 3.2. Characteristics of the Purified Glycoproteins

To further verify the properties of the purified LTIBTri-OPS_SPA_ and LTIIbBTri-OPS_SPA_, we analyzed their size, distribution, and stability using transmission electron microscopy (TEM) and dynamic light scattering (DLS). The TEM images illustrated that the size of the self-assembled LTIBTri-OPS_SPA_ was approximately 30 nm (Figure 2A). Subsequently, we named the LTIBTri-OPS_SPA_ particle as NP(LTIB)-OPS_SPA_. Moreover, the DLS analyses revealed a monodisperse and homogeneous distribution of the NP(LTIB)-OPS_SPA_ that agreed with the TEM observations (Figure 2B), signifying their ability to self-assemble into nanoparticles while being expressed. In addition, the stability of the particle was detected using DLS at different time points after incubation at 37 °C, showing a constant size distribution (Figure 2C). Correspondingly, we also analyzed LTIIbBTri-OPS_SPA_ and found that the size of the LTIIbBTri-OPS_SPA_ particle (renamed as NP(LTIIbB)-OPS_SPA_) was also approximately 30 nm (Figure 2D,E) and able to maintain stability at 37 °C for at least one week (Figure 2F). Therefore, similarly to CTBTri, the newly formed monomers (LTIBTri and LTIIbBTri) created by fusing Tri at the C-terminus of each pentamer, were also able to self-assemble into nanoparticles.

### 3.3. Safety Evaluation of the Four Types of Bioconjugate Nanovaccines

To evaluate the immune response and protective effects of the two nanoconjugate vaccines, we first assessed their safety in mice. The two vaccines, NP(LTIB)-OPS_SPA_ and NP(LTIIbB)-OPS_SPA_, were administered subcutaneously at 10 times the conventional immune dose, and safety parameters were recorded at different time points. In addition, untreated mice served as the control group (Figure 3A). During a 15-day observation period, the weight and temperature changes in the vaccinated mice were consistent with those in the control group, with no significant difference (Figure 3A,B). Moreover, we collected blood from the mice’s tails at different time intervals and analyzed the concentrations of the inflammatory factors, including IFN-γ, IL-1β, TNF-α, and IL-6, in the serum. Although a slight increase in some inflammatory factors was observed 3 days post-immunization, they remained within a very safe range, suggesting that none of the four vaccines triggered systemic inflammatory responses (Figure 3C). We then tested biochemical markers, including ALT, AST, ALP, BUN, and LDH, in the serums 30 days post-injection. The results demonstrated normal ranges of each index with no significant deviations from the control group (Figure 3D). Finally, we performed histological examinations on the major organs, including the hearts, livers, spleens, lungs, and kidneys. The hematoxylin and eosin (H&E) staining results indicated the absence of any organ damage (Figure 3E).

### 3.4. Evaluation of the Antibody Response and Protective Effect of the S. paratyphi A Bioconjugate Nanovaccines

Having confirmed the safety of both bioconjugate nanovaccines, we further determined their immune responses. In addition, PBS, OPS_SPA_, and the reported NP(CTB) (using the reported CTBTri as the carrier) were used as controls. After confirming the low endotoxin concentrations of the NP(LTIB)-OPS_SPA_ and NP(LTIIbB)-OPS_SPA_ (Appendix A), the BALB/c mice were immunized with one of the five treatments (PBS, OPS_SPA_, NP(CTB)-OPS_SPA_, NP(LTIB)-OPS_SPA_, and NP(LTIIbB)-OPS_SPA_) on days 0, 14, and 28. Blood was sampled 7 days after each immunization (days 7, 21, and 35) for the antibody analysis. On day 42, all of the mice were challenged with *S. paratyphi* A CMCC 50973, and mouse survival was monitored (Figure 4A). The ELISA-based measurements of the IgG titers against *S. paratyphi* A CMCC 50973 LPS in the serum samples revealed increases in all the NP(CTB)-OPS_SPA_, NP(LTIB)-OPS_SPA_, and NP(LTIIbB)-OPS_SPA_ immunizations, with the increases being most profound for the NP(CTB)-OPS_SPA_-, NP(LTIB)-OPS_SPA_-, and NP(LTIIbB)-OPS_SPA_-treated mice. However, the OPS_SPA_ barely produced a strong and long-lasting immune response (Figure 4B). By analyzing the four IgG subtypes from the last immunized serum, we found that IgG1 and IgG2a were predominant, and the titers in the NP(CTB)-OPS_SPA_, NP(LTIB)-OPS_SPA_, and NP(LTIIbB)-OPS_SPA_ groups were also higher than that in the other groups, which agreed with the total IgG results (Figure 4C). Further analysis of the IgG1-to-IgG2a ratio indicated that the immune response induced by NP(CTB)-OPS_SPA_ or NP(LTIIbB)-OPS_SPA_ maintained a Th1/Th2 balance (a ratio between one and two) (Figure 4D). On day 42, each mouse was intraperitoneally challenged with 1.89 × 10^8^ CFU of the *S. paratyphi* A CMCC 50973 strains. During observation, all of the mice died in the PBS group within 2 days, and only one mouse survived in the OPS_SPA_ group. The mice treated with NP(CTB)-OPS_SPA_ and NP(LTIB)-OPS_SPA_ received the best protection, exhibiting an survival of over 90% (Figure 4E).

### 3.5. Production of the LTB-Based Bioconjugate Nanovaccine against S. flexneri and K. pneumoniae

The above results demonstrated that LTB, as a pentamer module in the system, self-assembled into bioconjugate nanovaccines. In particular, LTIB may have performed the best. To determine the universality of this conclusion, we further coupled polysaccharides from the *S. flexneri* 2a strain 301 and the *K. pneumoniae* strain 355 on the two nanocarriers for further evaluation. The construction of the expression strains was performed as described above by introducing the plasmids pPglL-LTIBTri or pPglL-LTIIbBTri into the host strains 301DWP and 355DW, respectively. These two host strains have been constructed in the past, and the glycosylation system has been demonstrated to be functional [6,32]. After induction and purification through the same method as before, we successfully obtained the corresponding glycoproteins with purities greater than 90%. The protein components in the *S. flexneri* 2a strain 301 glycoproteins (including NP(LTIB)-OPS_Sf_ and NP(LTIIbB)-OPS_Sf_) and the *K. pneumoniae* strain 355 glycoproteins (including NP(LTIB)-OPS_Kp_ and NP(LTIIbB)-OPS_Kp_) were confirmed by Western blotting using an HRP-labeled 6×His Tag antibody (Figure 5A,B). Additionally, a specific serum against *S. flexneri* 2a strain 301 and *K. pneumoniae* strain 355 polysaccharides was used to determine their polysaccharide components. The results indicated that we successfully prepared various bioconjugate nanovaccines for two different pathogens.

### 3.6. Size and Distribution Analysis of the S. flexneri and K. pneumoniae Bioconjugate Nanovaccines

After successfully synthesizing four glycoproteins with different carriers from two pathogens, we analyzed the size and distribution of these glycoproteins using DLS. Although the results showed a slight decrease in particle size compared to those in *S. paratyphi* A in this study, due to the shorter polysaccharide chain lengths of the *S. flexneri* 2a strain 301 and the *K. pneumoniae* strain 355, all of the particles (including NP(LTIB)-OPS_Sf_, NP(LTIIbB)-OPS_Sf_, NP(LTIB)-OPS_Kp_, and NP(LTIIbB)-OPS_Kp_) remained in a size range between 25 and 30 nm with homogeneous size distributions (Figure 6A–D). In addition, during the seven-day observation period, all of the nanoparticles stably existed, indicating good stability (Figure 6E–H). Thus, the changes in the polysaccharide antigens did not affect the self-assembly of the nanoparticles.

### 3.7. Evaluation of the Immune Efficacy of LTBs-Based S. flexneri Bioconjugate Nanovaccines

The endotoxin concentrations of NP(LTIB)-OPS_Sf_ and NP(LTIIbB)-OPS_Sf_ were determined to be at a low level (Appendix A), and the immune effect of the *S. flexneri* bioconjugate nanovaccines (NP(LTIB)-OPS_Sf_ and NP(LTIIbB)-OPS_Sf_) were assessed using subsequent mouse experiments. Additionally, PBS, OPS_sf_, and NP(CTB)-OPS_Sf_ were used as controls. The procedure for immunization and evaluation, depicted in Figure 7A, aligns with the methodology for the *S. paratyphi* A vaccine evaluation. The ELISA results showcased a significant increase and highest levels in the IgG titer against the *S. flexneri* 2a strain 301 LPS in mice immunized with either NP(CTB)-OPS_Sf_ or NP(LTIB)-OPS_Sf_ (Figure 7B). Further analysis of the IgG subtypes in the final serum indicated that both the NP(CTB)-OPS_Sf_ and NP(LTIB)-OPS_Sf_ groups exhibited the highest titers, with a remarkable increase in IgG1, at least 100-fold in comparison to the OPS_Sf_ group (Figure 7C). Additionally, the IgG1-to-IgG2a ratio indicated that the immune response induced by NP(CTB)-OPS_Sf_, NP(LTIB)-OPS_Sf_, or NP(LTIIbB)-OPS_Sf_ also maintained a Th1/Th2 balance (a ratio between one and two) (Figure 7D). Subsequently, each mouse was intraperitoneally challenged with 1.53 × 10^7^ CFU of the wild-type *S. flexneri* 2a strain 301, and their survival was monitored (Figure 7E). During the observation period, all of the mice immunized with PBS died within 2 days, and the OPS_Sf_ and NP(LTIIbB)-OPS_Sf_ groups had very low protection rates (lower than 30%). Conversely, a 100% survival rate was noted in both the NP(CTB)-OPS_Sf_- and NP(LTIB)-OPS_Sf_-treated mice, indicating that CTB and LTIB, as pentamer modules, greatly enhanced the specific antibody response and increased the protection provided. These findings were consistent with those noted in the *S. paratyphi* A vaccine assessment.

### 3.8. Evaluation of the Immune Efficacy of the K. pneumoniae Bioconjugate Nanovaccines

After determining the concentrations of NP(CTB)-OPS_Kp_ and NP(LTIB)-OPS_Kp_ (Appendix A), we further explored the immune enhancement effects of the two nanocarriers for the polysaccharides derived from *K. pneumoniae* strain 355. Consistent with previous methodologies, we assessed the immune effect in mice (Figure 8A). On day 35, blood was drawn from each mouse, and the ELISA results showcasing the serum IgG antibodies against the *K. pneumoniae* strain 355 LPS echoed the trends observed in the two prior tests (Figure 8B). Notably, the mice treated with NP(CTB)-OPS_Kp_ and NP(LTIB)-OPS_Kp_ recorded the highest IgG titers. These mice were then injected intraperitoneally with 3.57 × 10^7^ CFU of the *K. pneumoniae* strain 355 (Figure 8C). It was distinctly observed that NP(CTB)-OPS_Kp_ and NP(LTIB)-OPS_Kp_ offered the highest protection rate (80%), while OPS_Kp_ provided a protection rate of only 20%. Thus, in agreement with past outcomes, we posit that CTB and LTIB, as pentamer modules, are more likely to elicit powerful specific immune responses and protection than other alternatives.

## 4. Discussion

In this study, we successfully prepared two LTB-based bioconjugate nanovaccines for three different pathogenic bacteria. The evaluation of the prophylactic effects against each pathogen revealed consistent results. LTIB, as a module, elicited higher immune responses than LTIIbB. Additionally, in a comparison with the reported self-assembling carrier NP(CTB), NP(LTIB) exhibited consistently good results. Thus, we have created a novel promising LTB-based bioconjugate of a self-assembled NP for vaccine design.

Many studies have indicated a significant correlation between the size of a vaccine and its immune efficacy. Specifically, nanoscale carriers can easily target lymph nodes, with the optimal size often ranging between 15 and 100 nm [15,33,34,35,36]. Our previous research has confirmed that conjugate vaccines sized 20–30 nm accumulated more readily in the lymph nodes and provoked efficient immune responses compared to non-nanovaccines. In this study, despite all of the vaccine sizes being essentially the same, we still observed substantial differences in immune efficacy, indicating that the size of the vaccine was not the only factor affecting the immunological efficacy.

Bacterial AB5 toxins perform various biological functions due to their diverse cell surface receptor recognition [29]. CTB is regarded as a mucosal adjuvant and is employed in oral vaccines [37]. During Vibrio cholerae infection, CTB can bind with GM1 on the surface of intestinal epithelial cells, mediating the entry of the toxin into the cells [38]. This might be the principle of CTB’s mucosal adjuvant effects. In addition, GM1 also resides on antigen-presenting cell (APC) surfaces, potentially promoting an immune response through GM1-mediated endocytosis. However, there are currently no particular studies on this mechanistic detail in immune cells. Similarly, LTIB also can bind with high affinity to GM1 [39]. In addition, in our results, NP(CTB)-OPS and NP(LTIB)-OPS showed consistent results, suggesting that GM1 may play an important role in the immune response. The best immune effects of both groups also indicated that GM1 may be a potential immune target for vaccine design. LTIIbB binds strongly to GD1a but lacks affinity for GM1 [39]. It has been reported that GD1 is also an essential coreceptor for TLR2 signaling [40].

Receptors can play a crucial role in the vaccine immune response, such as the activation of the immune response through Toll-like receptors (TLRs), C-type lectin receptors (CLRs), or the GAS-STING pathway [41], enhancing the APC’s presentation capability by influencing the fatty acid metabolism pathway [42]. Furthermore, some nanocarriers have been reported to activate the immune response through specific receptors (such as the binding of ferritin and CD209b on APCs) [43]. CTB, with its binding ability to GM1, is often used as a mucosal adjuvant, hence it can also be combined with mucosal delivery systems, such as recombinant lactic acid bacteria, to develop mucosal vaccines [44]. It should be emphasized that revealing the function of these receptors in vaccine immune responses would significantly advance the design of novel delivery vectors. In particular, CTB or LTIB modules can potentially elicit stronger immune responses, suggesting new adjuvant targets.

Although various B5 toxins can be used for bioconjugate nanovaccine design, the reported glycosylation efficiency and vaccine yield have been varied, possibly because the self-assembled structure with different B5 toxins affects the glycosylation reaction. Although the process of protein self-assembly and glycosylation is complex, there remains various methods to optimize the glycosylation efficiency and improve the yield. For example, by modifying the length or properties of the amino acid between two modules, it is possible to obtain a linker that is more suitable for self-assembly [24]. Furthermore, since the amino acid sequence of the glycosylation motif significantly impacts the modification efficiency [6], we can, therefore, mutate the key amino acids to obtain the highest-yielding strain. In addition to optimizing nanocarriers, modifications of the host chassis (such as the construction of engineering strains suitable for efficient glycoprotein synthesis) and the optimization of glycosyltransferases (such as the directed evolution and modification of the enzymes) can also improve vaccine yield [45].

## 5. Conclusions

In conclusion, we successfully prepared two LTB-based bioconjugate nanovaccines for various pathogenic bacteria. By evaluating the antibody response and protection against lethal bacterial challenges, we observed that the nanovaccine using LTIB as a module elicited a stronger immune response, with excellent results similar to the use of the previously reported CTBTri carrier. Thus, we have created the first novel promising LTB-based self-assembled NP for vaccine design. This NP has the potential for broad applications and expands the design of multifunctional self-assembled nanocarriers.

## Figures and Tables

**Figure 1 vaccines-12-00347-f001:**
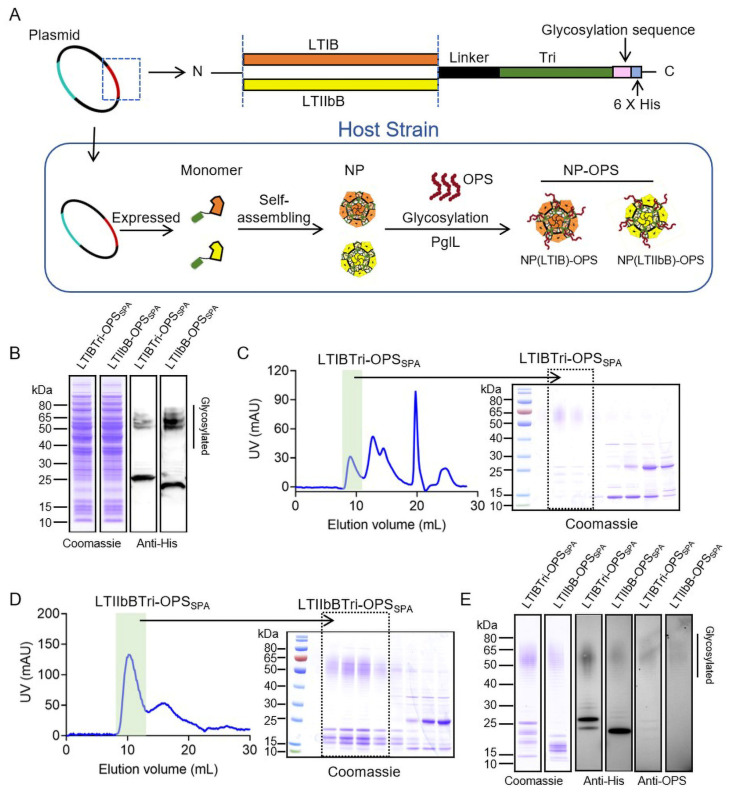
Expression and purification of glycoproteins. (**A**) Diagram of the nanovaccine expression process in host cells. (**B**) The plasmids pPglL-LTIBTri or pPglL-LTIIbBTri were transformed into the host strain 50973DWL/LT2, and then the expression of glycoproteins was detected by Coomassie blue staining and Western blotting using the HRP-labeled 6×His Tag antibody. When purified with a 24 mL column of size-exclusive chromatography (Superdex 200), LTIBTri-OPS_SPA_ (**C**) and LTIIbB-OPS_SPA_ (**D**) were analyzed from the collected fractions by Coomassie blue staining. (**E**) Analysis of the purified glycoproteins by Coomassie blue staining and Western blot using an HRP-labeled 6×His tag antibody and specific anti-OPS serum.

**Figure 2 vaccines-12-00347-f002:**
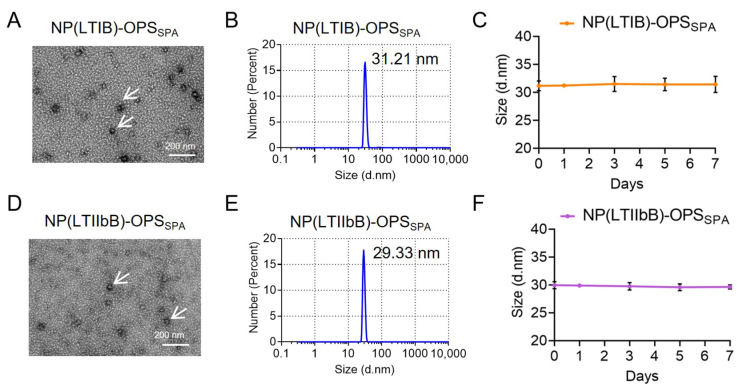
Characteristics of the purified *S. paratyphi* A glycoproteins. The purified glycoprotein LTIBTri-OPS_SPA_ was analyzed using TEM (**A**) and DLS (**B**). (**C**) After filtering with a 0.22 µm filter, the NP(LTIB)-OPS_SPA_ solution was incubated at 37 °C, and the stability was analyzed using DLS. (**D**) TEM image analysis of the LTIIbBTri-OPS_SPA_ particle (NP(LTIIbB)-OPS_SPA_). (**E**) NP(LTIIbB)-OPS_SPA_ was analyzed using DLS. (**F**) The stability of NP(LTIIbB)-OPS_SPA_ was analyzed using DLS at different time points. The arrow in the TEM image points to the nanovaccines.

**Figure 3 vaccines-12-00347-f003:**
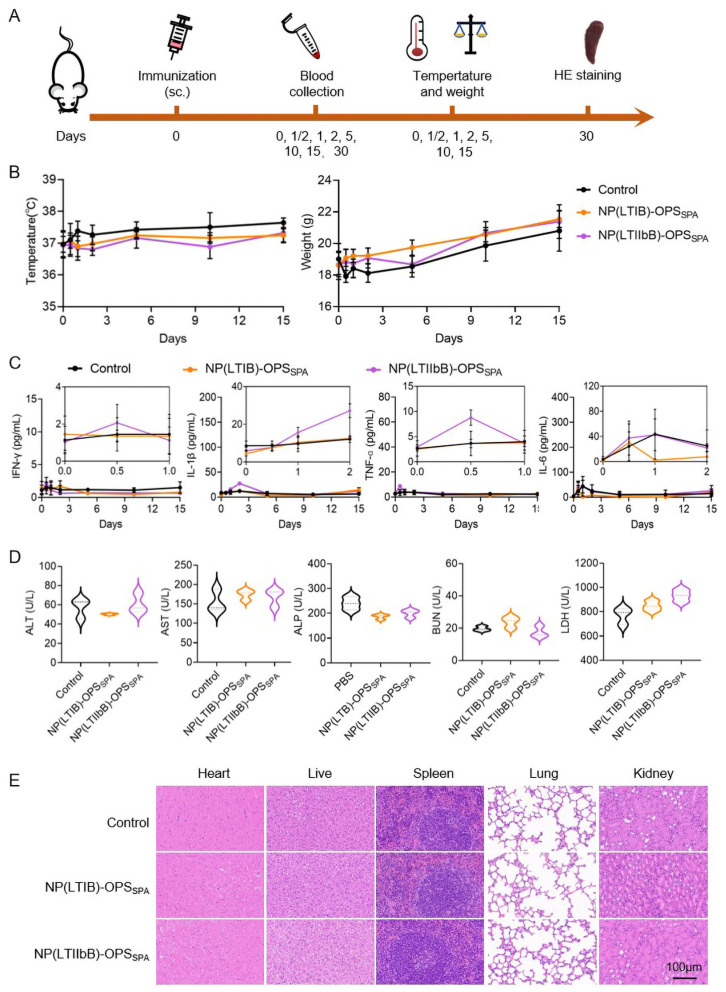
Safety evaluation of the two types of bioconjugate nanovaccines. (**A**) Schedule for safety evaluation experiments. (**B**) Temperature and weight changes in mice immunized with the two nanovaccines (containing 25 µg polysaccharide per mouse) during the 15-day observation period. (**C**) Serum cytokine profiles (IFN-γ, IL-1β, TNF-α, and IL-6) at different time points were measured after immunization. (**D**) Detection of the serum biochemical indices, including ALT, AST, ALP, BUN, and LDH, 30 days post-injection. (**E**) Representative images of the H&E staining of the hearts, livers, spleens, lungs, and kidneys from tissue sections at 30 days post-injection.

**Figure 4 vaccines-12-00347-f004:**
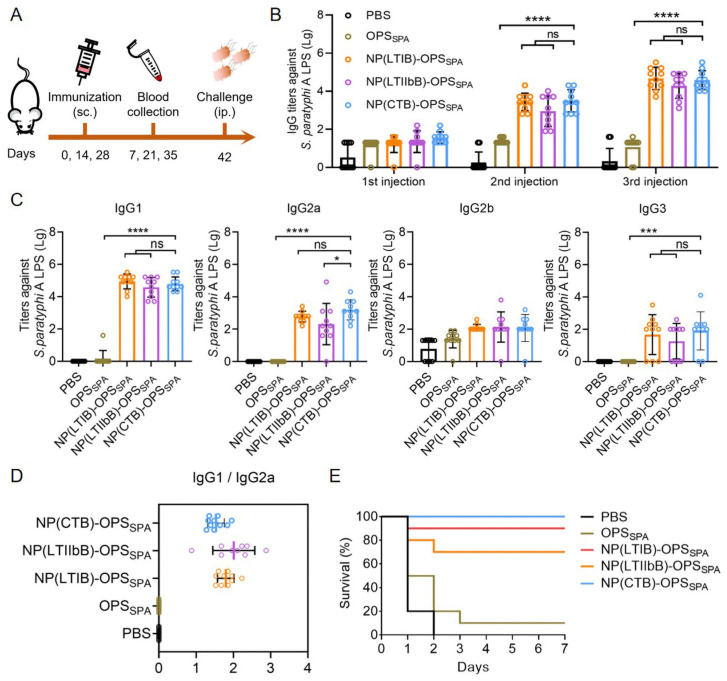
Evaluation of the antibody response and protective effect of the *S. paratyphi* A bioconjugate nanovaccines. (**A**) Schematic diagram of the immunization schedule. (**B**) IgG titers against *S. paratyphi* A CMCC 50973 LPS from the serums immunized with PBS, OPS_SPA_, NP(CTB)-OPS_SPA_, NP(LTIB)-OPS_SPA_, or NP(LTIIbB)-OPS_SPA_ 7 days after each injection (n = 10). (**C**) IgG subtype titers (IgG1, IgG2a, IgG2b, and IgG3) against *S. paratyphi* A CMCC 50973 LPS were measured from serum after the third injection (day 35) (n = 10). (**D**) Analysis of the rate of IgG1/IgG2a from the last immunized serum. (**E**) Mice survival was monitored after they were challenged with the *S. paratyphi* A CMCC 50973 strain (1.89 × 10^8^ CFU per mouse) 14 days after final the immunization (n = 10). Data are presented as means ± SD. Each group was compared using one-way ANOVA with Dunnett’s multiple-comparison test: **** indicates *p* < 0.0001, *** indicates *p* < 0.001, * indicates *p* < 0.05, and ns indicates *p* > 0.05.

**Figure 5 vaccines-12-00347-f005:**
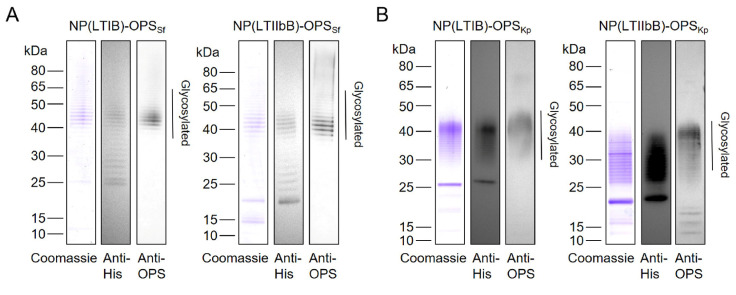
Expression and purification of the bioconjugate nanovaccines from the two hosts. (**A**) The plasmids pPglL-LTIBTri or pPglL-LTIIbBTri were transformed into the host strain 301DWP. After purification through affinity and size-exclusion chromatography as described before, glycoproteins, including NP(LTIB)-OPS_Sf_ and NP(LTIIbB)-OPS_Sf_, were obtained and analyzed by Coomassie blue staining and Western blotting using an HRP-labeled 6×His tag antibody and specific anti-OPS serum for detection. (**B**) The plasmids, pPglL-LTIBTri or pPglL-LTIIbBTri, were transformed into the host strain, 355DW. After purification through affinity and size-exclusion chromatography, glycoproteins NP(LTIB)-OPS_Kp_ and NP(LTIIbB)-OPS_Kp_ were obtained and analyzed by Coomassie blue staining and Western blotting using an HRP-labeled 6×His tag antibody and specific anti-OPS serum for detection.

**Figure 6 vaccines-12-00347-f006:**
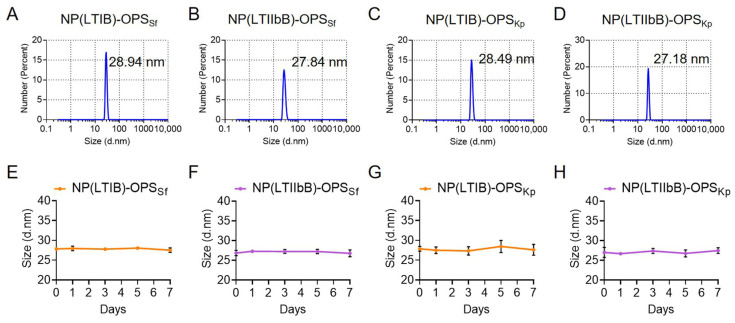
Analyses of the purified glycoproteins. (**A**–**D**) The four purified glycoproteins, namely, NP(LTIB)-OPS_Sf_, NP(LTIIbB)-OPS_Sf_, NP(LTIB)-OPS_Kp_, and NP(LTIIbB)-OPS_Kp,_ were analyzed using DLS. (**E**–**H**) The stability of the four nanovaccines was analyzed. After filtering with a 0.22 µm filter, the NP(LTIB)-OPS_Sf_, NP(LTIIbB)-OPS_Sf_, NP(LTIB)-OPS_Kp_, and NP(LTIIbB)-OPS_Kp_ solutions were incubated at 37 °C, and the sizes were detected using DLS at different time points.

**Figure 7 vaccines-12-00347-f007:**
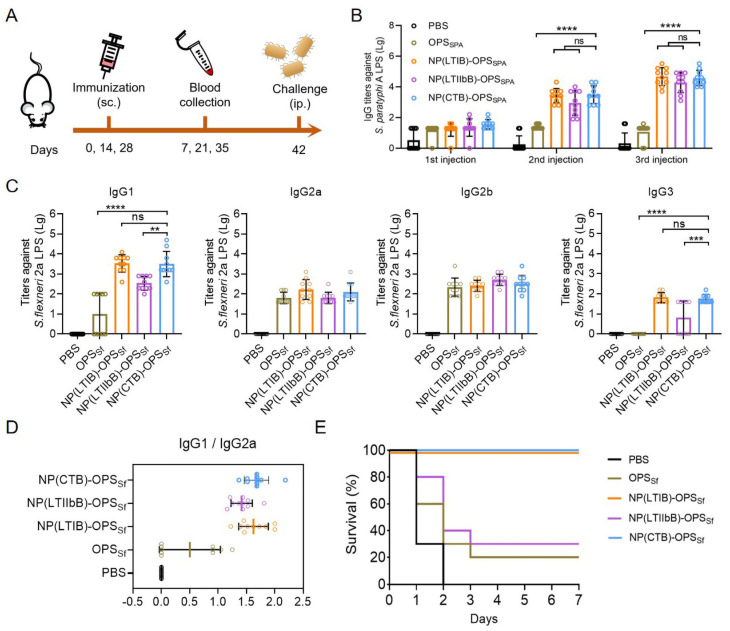
Evaluation of the antibody response and protective effect of the *S. flexneri* bioconjugate nanovaccines. (**A**) Schematic diagram of the immunization schedule. (**B**) IgG titers against the *S. flexneri* 2a strain 301 LPS in the serum. BALB/c mice were immunized with PBS, OPS_Sf_, NP(CTB)-OPS_Sf_, NP(LTIB)-OPS_Sf_, or NP(LTIIbB)-OPS_Sf_, and the serums were sampled 7 days after each injection (n = 10). (**C**) IgG subtype titers (IgG1, IgG2a, IgG2b, and IgG3) against the *S. flexneri* 2a 301 strain LPS were measured in the serum after the third injection (day 35) (n = 10). (**D**) Analysis of the rate of IgG1/IgG2a from the last immunized serum. (**E**) Mice survival was monitored after they were challenged with the *S. flexneri* 2a strain 301 (1.53 × 10^7^ CFU per mouse) 14 days after the final immunization (n = 10). Data are presented as means ± SD. Each group was compared using one-way ANOVA with Dunnett’s multiple-comparison test: **** indicates *p* < 0.0001, *** indicates *p* < 0.001, ** indicates *p* < 0.01, and ns indicates *p* > 0.05.

**Figure 8 vaccines-12-00347-f008:**
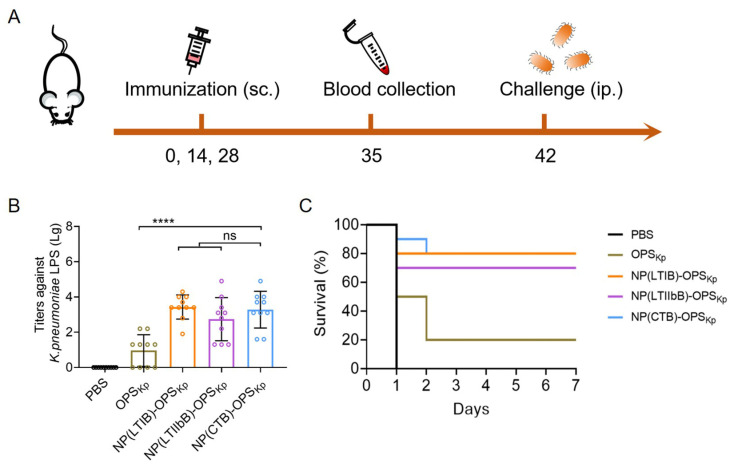
Evaluation of the antibody response and protective effect of the *K. pneumoniae* bioconjugate nanovaccines. (**A**) Schematic diagram of the immunization schedule. (**B**) IgG titers against the *K. pneumoniae* strain 355 LPS in the serum. BALB/c mice were immunized with PBS, OPS_Kp_, NP(CTB)-OPS_Kp_, NP(LTIB)-OPS_Kp_, and NP(LTIIbB)-OPS_Kp_, and the serums were sampled 7 days after the third injection (day 35) (n = 10). (**C**) At 14 days after the final immunization, each mouse was challenged with 3.57 × 10^7^ CFU of the *K. pneumoniae* strain 355, and the survival of the mice was monitored (n = 10). Data are presented as means ± SD. Each group was compared using one-way ANOVA with Dunnett’s multiple-comparison test: **** indicates *p* < 0.0001 and ns indicates *p* > 0.05.

## Data Availability

The data presented in this study are available on request from the corresponding author.

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
