# Peer review of "Production of Promising Heat-Labile Enterotoxin (LT) B Subunit-Based Self-Assembled Bioconjugate Nanovaccines against Infectious Diseases"

_vaccines, 2024, doi:10.3390/vaccines12040347_

Round 1

Reviewer 1 Report

Comments and Suggestions for Authors

In this manuscript, the authors developed a heat-labile protein-based nanoparticle platform that displays bacterial O-polysaccharides as vaccines. The purified particles were evaluated for size, stability, and safety in the mouse model. In addition, the nanoparticle was applied to display the OPS from 3 different bacteria strains and conferred good IgG-mediated protection in mice. Please address the following comments.

1.     The SEC curves and the bands on the gels indicate the purity is relatively low. Please provide evidence to prove that the nanoparticles were not contaminated with LPS originating from the bacterial lysate.

2.     Have you performed any analysis to identify the copy number of LTB-Tri monomers in each particle? Does the nanoparticle have icosahedral symmetry? Can you make a 3D reconstruction model of the nanoparticles based on TEM data?

3.     In the safety evaluation study, did you observe any lesions or signs of adverse effects around the injection site?

4.     One would expect changes in the levels of inflammatory factors after injection of foreign antigens. Could you please explain why you did see that in Figure 3C? Also, I would suggest rescaling the Y axis in Figure 3C, the scales in the zoomed-in figures in squares seem to be appropriate.

5.     I’m not a glycan expert. I’m curious how different, in terms of structure, the OPS from Spa/Ssf/Skp are. How did you purify the unconjugated OPS (for immunization) or LPS (for ELISA) for different bacteria? It’s not very clear how OPS from different bacteria were conjugated to the nanoparticle, are the glycosylation patterns on the LTB_Tri the same as the glycans on the bacterial surface? Could you please provide more details about this?

Figure 1. Please specify which lane is from which SEC fraction in the Coomassie staining of C and D.

Line 225: please rephrase here “Subsequently, we named LTIBTri-OPSSPA particle as NP(LTIB)-OPSSPA”

Comments on the Quality of English Language

There are some grammar errors and some sentences that need polishing. Please carefully proofread the manuscript. 

Reviewer 2 Report

Comments and Suggestions for Authors

The authors developed LTB-containing self-assembled Nanoparticle vaccines and evaluated them in mice. They found NP(LTIB) vaccine can induced high antibody titers and protect mice from later challenge. The idea of using glycosylation system is unique and promising, however the advantage over the current method is not clear. Comments for the authors below:

Major points:

1.     Line 150: Please indicate the injection volume and diluent.

2.     Line 160: Please indicate the antigen and adjuvant dose and volume per mouse.

3.     Figure 1A: Please indicate the His-tag position.

4.     Have the authors consider to measure anti-Tri antibodies.

5.     Please explain why the authors did not test intranasal or oral immunization routes by considering LTB is a mucosal adjuvant.

Minor points:

1.     Line 107: Please add a reference for previous work.

2.     Line 172: “nmol” should be “mM”.

Round 2

Reviewer 1 Report

Comments and Suggestions for Authors

All previous comments are well addressed.